# Expression of Functional Cannabinoid Type-1 (CB_1_) Receptor in Mitochondria of White Adipocytes

**DOI:** 10.3390/cells11162582

**Published:** 2022-08-19

**Authors:** Antonio C. Pagano Zottola, Ilenia Severi, Astrid Cannich, Philippe Ciofi, Daniela Cota, Giovanni Marsicano, Antonio Giordano, Luigi Bellocchio

**Affiliations:** 1INSERM U1215 Neurocentre Magendie, Université de Bordeaux, 33077 Bordeaux, France; 2Department of Experimental and Clinical Medicine, Università Politecnica delle Marche, 60121 Ancona, Italy

**Keywords:** CB_1_ receptor, mitochondria, white adipose tissue

## Abstract

Via activation of the cannabinoid type-1 (CB_1_) receptor, endogenous and exogenous cannabinoids modulate important biochemical and cellular processes in adipocytes. Several pieces of evidence suggest that alterations of mitochondrial physiology might be a possible mechanism underlying cannabinoids’ effects on adipocyte biology. Many reports suggest the presence of CB_1_ receptor mRNA in both white and brown adipose tissue, but the detailed subcellular localization of CB_1_ protein in adipose cells has so far been scarcely addressed. In this study, we show the presence of the functional CB_1_ receptor at different subcellular locations of adipocytes from epididymal white adipose tissue (eWAT) depots. We observed that CB_1_ is located at different subcellular levels, including the plasma membrane and in close association with mitochondria (mtCB_1_). Functional analysis in tissue homogenates and isolated mitochondria allowed us to reveal that cannabinoids negatively regulate complex-I-dependent oxygen consumption in eWAT. This effect requires mtCB_1_ activation and consequent regulation of the intramitochondrial cAMP-PKA pathway. Thus, CB_1_ receptors are functionally present at the mitochondrial level in eWAT adipocytes, adding another possible mechanism for peripheral regulation of energy metabolism.

## 1. Introduction

The endocannabinoid system (ECS) is an important modulator of food intake and energy balance [1]. Cannabinoid type-1 (CB_1_) receptors and their main endogenous lipid ligands, 2-arachidonoyl-glycerol and anandamide, are largely present in the brain and in peripheral organs involved in the regulation of energy metabolism, such as the liver, skeletal muscle, pancreas, gastrointestinal tract, and adipose tissue [1]. Pharmacological stimulation of CB_1_ receptors generally leads to an increase in energy intake and storage, whereas CB_1_ antagonists exert the opposite effects in both animals and humans [2]. Pre-clinical and clinical data show a close association between obesity and a chronic pathological overactivation of ECS, as indicated by an overproduction of endocannabinoids and overexpression of CB_1_ receptors [3]. This phenomenon has been described at both the central and peripheral level, and especially in adipose tissue [4].

The anatomical localization of fat depots, in association to the metabolic status of adipocytes, determines the classification of adipose tissues into white (WAT) or brown (BAT) [5]. Whereas BAT has a crucial role in adaptive thermogenesis, WAT is important for the storage of energy in the form of triglycerides and is a source of a number of endocrine signals [6]. Mitochondrial activity and its products, such as reactive oxygen species (ROS), play a crucial role not only in thermogenesis and adipocyte characterization [7] but also in the homeostasis of WAT by directly regulating adipo-lipogenesis, lipolysis, fatty acids, and ketone body metabolism [7,8]. Accordingly, pharmacological and genetic interventions that modulate the mitochondrial functions of adipose tissue impact on energy metabolism and affect the development of obesity and associated metabolic disorders, such as type 2 diabetes and the accompanying insulin resistance [7,8].

Extensive in vitro work has shown that direct CB_1_ receptor activation in adipocytes is able to stimulate cell growth and cell differentiation, to alter adipokines secretion and stimulate lipogenesis, resulting in adipocytes that contain higher amounts of lipids [9,10,11,12]. Furthermore, the CB_1_ receptor negatively regulates mitochondrial biogenesis in WAT, resulting in reduced mitochondrial mass [13]. Last but not least, body weight loss induced by chronic administration of CB_1_ receptor antagonists is largely due to increased energy expenditure and consequent activation of lipolysis and fatty acid oxidation in adipocytes [9,14,15,16]. Thus, mitochondrial activity appears to be a key target of CB_1_-mediated alteration of adipose physiology.

Even expressed at very low levels, mitochondrial-associated CB_1_ receptor (mtCB_1_) has been described in several cell types, including brain cells, muscle, sperm, and oocytes [17,18,19,20,21,22,23,24]. Activation of mitochondrial-associated CB_1_ receptors (mtCB_1_) directly regulates the activity of these organelles. This regulation has been shown to exert several biochemical effects in neurons and astrocytes, including ATP production, modulation of ROS, and neuropeptide signaling, determining crucial behavioral readouts that are associated to cannabinoids consumption [18,20,21]. Given the aforementioned role of mitochondria in adipose tissue physiology [7,8], we investigated whether adipocyte mitochondria possess a functional CB_1_ receptor in order to pave the way to understanding the mechanism through which endocannabinoids regulate adipocyte physiology.

## 2. Materials and Methods

### 2.1. Animal and Drugs

Mice and rats were maintained under standard conditions (food and water ad libitum; 12 h–12 h light–dark cycle, light on at 7:00; experiments were performed between 9:00 and 17:00). Wild-type and *CB_1_*-KO male mice (2–4 months old) were obtained, bred, and genotyped as described [25]. Ati-*CB_1_*-KO mice were induced with tamoxifen 3 weeks before the experiments as previously described [16]. DN22-*CB_1_*-KI (lacking the mtCB_1_ receptor) mice were bred, genotyped, and maintained as described [22]. Wistar rats were purchased from Janvier (Marseille, FRANCE) and sacrificed for the experiment at the age of 12–16 weeks. Animal studies were approved by the Institutional Ethics Committee for the Care and Use of Experimental Animals of the University of Bordeaux, the Committee on Animal Health and Care of INSERM, and the French Ministry of Agriculture and Forestry (authorization number 3306369). All experiments were performed in accordance with the guidelines for animal use specified by the European Union Council Directive of 22 September 2010 (2010/63/EU) and approved by the French Ministry of Higher Education, Research, and Innovation (authorization number 3306369 and 20053).

WIN55,212-2 (WIN) and KH7 were purchased from Sigma Aldrich (Saint-Quentin-Fallavier, France). JD5037 was obtained from MedKoo biosciences (Morrisville, NC, USA). 8-Br-cAmp was obtained from Bio-Techne (Saint-Quentin-Fallavier, France). All the drugs were dissolved in DMSO apart for 8-Br-cAmp, which was re-suspended in distilled water.

### 2.2. Sample Preparation for Morphological Studies

Anesthetized mice were perfused transcardially with a fixative solution containing 4% paraformaldehyde and 0.05% glutaraldehyde in 0.1 M phosphate buffer (PB), pH 7.4. Brains, epididymal white (eWAT), and interscapular brown (iBAT) adipose tissue depots were collected, postfixed in the same fixative solution for 12 h at 4 °C, and washed in PB. Free-floating brain coronal sections (40 µm thick) were obtained with a Leica VT1200S vibratome (Leica Microsystems, Vienna, Austria) and kept in phosphate-buffered saline (PBS), pH 7.4, at 4 °C until use. Then, small eWAT specimens sized about 2 mm were obtained from each depot and processed for electron microscopy (see below), whereas the remaining tissue was dehydrated with ethanol, cleared with xylene, and embedded in paraffin.

### 2.3. Peroxidase Immunohistochemistry

Immunohistochemistry was performed on 3 µm thick paraffin-embedded sections of eWAT and iBAT from WT and *CB_1_*-KO mice. After a thorough rinse in phosphate-buffered saline (PBS), sections were reacted with 0.3% H_2_O_2_ (in PBS; 30 min) to block endogenous peroxidase, rinsed in PBS, and incubated in a 4% blocking solution (in PBS; 60 min). Then, they were incubated with a polyclonal goat anti-CB_1_ receptor primary antibody (Frontier Institute, Hokkaido, Japan, # CB_1_-Go-Af450; dilution at 1:40 *v/v* in PBS, overnight at 4 °C). After a thorough rinse in PBS, sections were incubated in a 1:200 *v/v* biotinylated anti-goat IgG secondary antibody (in PBS; 30 min). Histochemical reactions were performed using a Vectastain ABC kit (Vector Laboratories, Burlingame, CA, USA) and Sigma Fast 3,3′-diaminobenzidine (Merk, St. Louis, MO, USA) as chromogen. Sections were finally counterstained with hematoxylin, dehydrated, and mounted with Eukitt mounting medium. Staining was never observed when the primary antibody was omitted.

As positive/negative controls, hippocampal sections from WT and *CB_1_*-KO mice were subjected to the same type of analyses. Immunohistochemical detection of CB_1_ in the mouse hippocampus was performed on free-floating brain coronal sections applying the same protocol (dilution of the primary antibody at 1:200 *v/v*). Strong CB_1_ immunoreactivity was observed in the pyramidal cell layer and in the stratum radiatum of WT mice as previously described [26]. On the other hand, only a slight background labeling was observed in *CB_1_*-KO sections, confirming the sensitivity and specificity of the antibody used (Appendix A).

### 2.4. Immunogold Post-Embedding Technique

eWAT samples from WT and *CB_1_*-KO mice were processed according to an osmium-free embedding method [27]. Briefly, dehydrated small tissue fragments were immersed in propylene oxide, infiltrated with a mixture of Epon/Spurr resins, and polymerized at 60 °C for 48 h. After polymerization, samples were sectioned (thin sections of 60–70 nm) with an MT-X ultramicrotome (RMC, Tucson, AZ, USA). Thin sections were then mounted on 200 mesh nickel grids and processed for immunogold labeling as previously described [27]. In brief, after treatment with 4% p-phenylenediamine in Tris-buffered saline (0.1 M Tris, pH 7.6, with 0.005% Tergitol N P-10 (TBST)), grids were washed in TBST (pH 7.6), transferred for 15 min in 2% normal serum in TBST (pH 7.6), and then incubated overnight in a solution of TBST (pH 7.6) containing the polyclonal goat anti-CB_1_ receptor primary antibody (Frontier Institute, Hokkaido, Japan, # CB_1_-Go-Af450; dilution at 1:20 *v/v*). Grids were subsequently washed in TBST (pH 8.2), transferred for 10 min in 2% normal serum in TBST (pH 8.2), incubated for 2 h in TBST (pH 8.2) containing 18 nm gold particle-linked IgG anti-goat IgG (Jackson ImmunoResearch, West Grove, PA, USA, # 705-215-147, dilution at 1:20 *v/v*), washed in distilled water, and then stained with uranyl acetate and lead citrate. Labeled sections were examined with a CM10 transmission electron microscope (Philips, Eindhoven, Netherlands). The concentration of the primary antibody was determined by testing several dilutions. The concentration yielding the lowest level of background, calculated by estimating the gold density over adipocyte nuclei, was used to perform the final studies [28]. Gold particles were not detected when the primary antiserum was omitted.

### 2.5. Morphometric Analysis

Morphometric evaluations were performed on 6 WT and 6 *CB_1_*-KO mice (according to [17,18,19]). Three sections were analyzed for each animal. From each section, random electron micrographs were taken to study the morphology and immunoreactivity of the adipocytes. Mitochondrial labeling was considered positive when at least one immunogold particle was over the organelle or within approximately 25 nm from its outer membrane. The percentage of CB_1_-labeled mitochondria over the total number of mitochondria was calculated and normalized for 100 mitochondria for each sample. The density of mitochondrial CB_1_ labeling was calculated as the number of gold particles per area (µm^2^) of the outer membrane positive mitochondria. All the analyses were performed in a blind manner. A total of 520 mitochondria in WT and 613 mitochondria in *CB_1_*-KO were analyzed.

### 2.6. Homogenate Preparation of eWAT and iBAT Mouse Tissue

WT, *CB_1_*-KO, DN22-*CB_1_*-KI, and Ati-*CB_1_*-KO mice were sacrificed by cervical dislocation and eWAT and iBAT were rapidly dissected. The tissues were washed in ice-cold phosphate-buffered saline and homogenized in 600 µL of Mir05 without Taurin [29] using a polyron homogenizer. The sample was centrifuged twice at 500× *g* 4 °C for 1 min, collecting the infranatant fraction at each step. Digitonin was then added at the final concentration of 2 µM and incubated for 5 min. The obtained samples were then used for complex I assay or mitochondrial respiration.

### 2.7. Complex I Assay Activity

eWAT and iBAT homogenates were separated into 2 aliquots and then treated with WIN (5 µM) or vehicle for 10 min at 37 °C and immediately frozen on dry ice. Samples were kept at −80 °C until the enzymatic assay. Complex I activity was measured by recording the decrease in absorbance due to oxidation of NADH at 340 nm (ε = 6.2 mM^−1^ cm^−1^) in a POLARstar Omega plate reader (BMG Labtech, Ortenberg, Germany) at 30 °C using multi-well plates (12 well). In total, 100 µL of sample was then added to the assay buffer (44 mM K2HPO4 pH 7.2; 5.2 mM MgCl2, 2.6 mg/mL BSA, Antimycine 2 µg/mL, KCN 0.2 mM, 0.1 mM Decylubiquinone) and 0.1 mM NADH was added just before the measurement. The changes in the 340 nm absorbance (slope per minute) for the 2 conditions were recorded simultaneously for 3 min. The Complex I inhibitor rotenone was then added at the final concentration of 5 µg/mL, each sample was incubated for 5 min at 30 °C, and the absorbance recorded for a further 3 min. The protein concentration of eWAT homogenates was determined using the Roti-Nanoquant protein quantification assay (Carl Roth, Karlsruhe, Germany). Specific complex I activity was calculated as nmol min/mg protein using the following formula: (slope per minute × mL of reaction)/((ε × volume of samples in mL) × (sample protein concentration in mg/mL)). Enzyme activity was corrected for rotenone-resistant activity and the assay performed in duplicates per mouse sample. The WIN effect is reported as % of the vehicle condition.

### 2.8. Mitochondrial Isolation from eWAT Rat Tissue

Anesthetized rats were sacrificed by decapitation and eWAT was dissected, washed in cold phosphate-buffered saline, and then homogenized in mitochondrial isolation buffer (210 mM mannitol, 70 mM sucrose, 1 mM EDTA, 50 mM Tris pH 7.4) supplemented with protease inhibitors (Sigma-Aldrich, Saint-Quentin-Fallavier, France), using a politron homogenizer. In order to isolate the mitochondrial fraction, a series of centrifugations at 4 °C was carried out [30]. A first step at 500× *g* for 1 min was repeated 3 times, recovering the infranatant. The soluble fraction was then centrifuged at 1000× *g* for 10 min and the supernatant additionally spun down for 20 min at 7000× *g*. The resulting pellet was resuspended in mitochondrial isolation buffer and used for the respirometric assay or frozen to be processed for western blotting.

### 2.9. Mitochondrial Respiration

70 µL of cell lysate or 40 µL of mitochondrial preparation was loaded in each chamber of a 2K Oroboros device [29] together with malate (2 mM), pyruvate (5 mM), and glutamate (10 mM) (MPG) followed by 1.25 mM ADP to reach a stable OXPHOS state of respiration. Vehicle or WIN at the final concentrations of 5 µM was added to the chamber and the values of the oxygen consumption rate (OCR) was noted for 5 min. Subsequently, oligomycin (2.5 µM) and carbonyl cyanide m-chlorophenylhydrazone (CCCP) (3 steps of 1 µM each) were injected in the chamber. The final values were ROX-corrected, adding rotenone (0.5 µM) and antimycin A (2.5 µM), and expressed as a percentage of the OXPHOS state. To perform a complete analysis of the quality of mitochondrial preparation, after ADP injection, cytochrome C (10 µM), succinate (10 mM), rotenone (0.5 µM), and subsequently Antimycin A (2.5 µM) were added [29]. For eWAT rat mitochondria, JD5037 (CB_1_ antagonist), KH7 (sAC inhibitor), and 8-br-cAMP (PKA activator) were applied as pretreatments at the final concentration of 2, 5, and 500 µM, respectively. The effects of WIN administration on mitochondrial respiration were calculated as changes over the OXPHOS state and normalized in % to the changes observed upon vehicle administration in order to compare the effect of CB_1_ activation on OCR in different genotypes or upon distinct pre-treatments [18].

### 2.10. Western Blotting

The protein concentration was determined using the Roti-Nanoquant protein quantification assay (Carl Roth, Karlsruhe, Germany) and the extract mixed with Laemmli loading buffer. For each condition, one sample aliquot was incubated at 37 °C for 30 min for CB_1_ and Gi immunoblotting while the rest was boiled for 5 min at 95 °C to immunoblot the other cellular markers. In total, 25 µg of protein per well was loaded on 4–20% precast polyacrylamide gels (Bio-Rad, Hercules, CA, USA) and transferred to 0.45-mm PVDF membranes (Merk Millipore, Billerica, MA). Membranes were immersed in a mixture of Tris-buffered saline and polysorbate 20 (20 mM Tris-HCl pH 7.6, 150 mM NaCl, 0.05% Tween 20) containing 5% non-fat milk for 1 h at room temperature. The antibodies used were: anti-CB_1_ (ab23703, Abcam, Cambridge, UK), anti-Gi (anti-Gi proteins alpha inhibitor 1 + 2, ab3522, Abcam, Cambridge, UK), anti-sAC (ADCY10, PA543049, Invitrogen, Carlsbad, CA, USA), PKA catalytic subunit (anti-cAMP Protein Kinase Catalytic subunit antibody, ab76238, Cambridge, UK), NDUFS4 (anti-NDUFS4 ab87399, Abcam, Cambridge, UK), anti-Glut1 (sc-7903, Santa Cruz Biotechnology, Dallas, TX, USA), anti-Rab7A (840401, Biolegend, San Diego, CA, USA), anti-LAMP-2 (sc-18822, Santa Cruz Biotechnology, Dallas, TX, USA), and anti-Tubulin (sc-69969, Santa Cruz Biotechnology, Dallas, TX, USA). Primary antibodies were detected with HRP-linked secondary antibodies purchased from Cell Signaling Technology (Danvers, MA, USA). Signal was detected by chemiluminescence detection (Clarity Western ECL Substrate, Bio-Rad, Hercules, California or Super Signal West Femto Maximum Sensivity Substrate, Thermo Fisher Scientific, Waltham, MA) and analyzed using the Image Lab software (Bio-Rad, Hercules, CA, USA) after acquisition on ChemiDoc Touch (Bio-Rad, Hercules, CA, USA).

### 2.11. Statistical Analysis

All graphic representation and statistical analyses of the data were performed using GraphPad software (version 8.0, San Diego, CA, USA). Results were expressed as means ± s.e.m. Data were analyzed using the appropriate statistic test as reported in Appendix A. Significances were expressed as follows: * *p* < 0.05, ** *p* < 0.01.

## 3. Results

### 3.1. Immunohistochemical Detection of CB_1_ in Adipose Tissue

Several reports suggest that CB_1_ receptor mRNA is expressed in both white and brown adipose tissue depots [16,31]. However, CB_1_ protein has not always been specifically detected, especially in iBAT [32,33,34,35]. This is likely due to differences in the age, sex, and metabolic status of the animals [32], and to the technical constrains of western immunoblotting or the immunohistochemical procedures.

To successfully characterize the intracellular localization of CB_1_ receptor by immunogold staining coupled to electron microscopy, we first assessed the expression and distribution of CB_1_ receptor by immunohistochemical staining on paraffin-embedded sections from eWAT. White adipocytes from eWAT of lean animals exhibited a slight but specific CB_1_ staining that was distributed over their thin cytoplasmic rim (Figure 1A). Notably, such staining was absent in the white adipocytes from the eWAT of *CB_1_*-KO mice (Figure 1A). On the other hand, iBAT sections from lean WT mice did not reveal any specific CB_1_ staining in brown adipocytes (Appendix A). These data indicate that eWAT adipocytes express low but detectable amounts of this receptor.

### 3.2. CB_1_ Receptor Is Present on Plasma Membrane and Associated to Nucleus and Mitochondria in eWAT Adipocytes

In order to resolve the intracellular distribution of CB_1_ receptors in eWAT adipocytes, we resorted to the immunogold post-embedding technique [27,28]. eWAT adipocytes from WT mice showed some CB_1_ gold particles distributed on the plasma membrane, in the nucleus and in the cytoplasm; some gold particles were also observed in the mitochondria (Figure 1B). In contrast, CB_1_ immunogold staining of mice bearing genetic CB_1_ deletion (*CB_1_*-KO) presented only background levels of immunogold staining (Figure 1B).

Morphometric analyses revealed a much higher number of gold particles in WT sections when compared to *CB_1_*-KO-derived samples, indicating the specificity of the staining (Figure 1B,C). Interestingly, the WT eWAT sections showed a consistently higher number of gold particles located on the adipocyte plasma membrane (Figure 1B,D) and in the nucleus (Figure 1B,E), with no change in the cytosolic compartment (Figure 1B,F) when compared to *CB_1_*-KO.

At the mitochondrial level, WT sections showed a much higher number of gold particles compared to the *CB_1_*-KO littermate samples (Figure 1B,G). Surprisingly, half of these gold particles were associated to external mitochondrial membranes (Mt-external CB_1_) (Figure 1B,H) while the rest were located inside mitochondria that are frequently close to the inner mitochondrial membrane (called “Mt-inside CB_1_”) (Figure 1B,H). We then analyzed the relative number of CB_1_-positive mitochondria. Despite similar total amounts of these organelles between the WT and *CB_1_*-KO samples (Figure 1B,I), the WT eWAT sections presented a significantly higher percentage of CB_1_-positive mitochondria (Figure 1B,J).

Collectively, these results show that mitochondria from eWAT adipocytes are specifically endowed with CB_1_ receptors.

### 3.3. CB_1_ Receptor Activation Impacts Mitochondrial Respiration in eWAT

Natural and synthetic agonists of CB_1_, such as Δ^9^-tetrahydro-cannabinol (THC) or WIN55,212-2 (WIN), reduce mitochondrial oxygen consumption by acting at mtCB_1_ [18,36,37]. We previously showed that this effect is mediated by the inhibition of the phosphorylation of complex I subunits [18]. Conversely, cannabinoids have no effect on oxygen consumption when this is mediated by complex II [18].

To study the role of the mitochondrial pool of CB_1_ receptor stimulation in adipose tissue, we treated eWAT and iBAT mouse homogenates with the cannabinoid agonist WIN at a concentration of 5 µM. The enzymatic assay for complex I activity in eWAT clearly showed a net decrease in WIN-treated samples from WT mice (Figure 2A). Strikingly, this effect was absent in samples derived from full *CB_1_*-KO and from DN22-*CB_1_*-KI (a mutant mouse line that lacks selective mitochondrial CB_1_ receptor localization [22]) (Figure 2A). Conversely, in iBAT, WIN treatment did not exert any effect in all genotype conditions (Appendix A).

To further characterize the effects of CB_1_ receptor activation on mitochondrial activity in adipocytes, we performed high-resolution respirometry analysis on mouse whole homogenate of eWAT using the Oroboros System [29]. With this system, we were able to constantly monitor mitochondrial integrity and isolate the contribution of the first two respiratory chain complexes using different substrates/inhibitors (Figure 2B).

In eWAT homogenates, WIN reduced complex-I-dependent mitochondrial respiration in WT-derived samples (Figure 2C,G) but not in the ones obtained from the *CB_1_*-KO littermates (Figure 2D,G), indicating the specificity of the CB_1_ receptor in this effect. Epididymal WAT is composed of a heterogeneous population of cells that includes fibroblasts, adipocyte precursors, endothelial cells, immune cells, and adipocytes, which all contain mitochondria [38]. To test whether the effects on mitochondrial respiration are due to the activation of mtCB_1_ receptors located in other cellular populations than adipocytes [5], we performed the same experiment in Ati-*CB_1_*-KO mice, which lack CB_1_ exclusively in fat cells [16] and WT littermates. The WIN effect was completely abolished in these mutant animals (Figure 2E,G), suggesting an adipocyte-specific effect. Furthermore, WIN also did not impact complex-I-dependent mitochondrial respiration in homogenates obtained from DN22-*CB_1_*-KI mice (which lacks mitochondrial CB_1_ receptor localization) [22] (Figure 2F,G), in agreement with previous findings in brain tissues [22].

Altogether, these data indicate that the inhibitory effect of cannabinoids is selective to eWAT mitochondrial activity and is mainly due to adipocyte mtCB_1_ activation.

### 3.4. mtCB_1_ Reduces Mitochondrial Respiration in eWAT via sAC and PKA Activity

Previous evidence showed that mtCB_1_ activation in the brain inhibits soluble adenylyl cyclase (sAC) and reduces the intramitochondrial levels of cAMP, resulting in decreased PKA-dependent complex I phosphorylation and lowered mitochondrial respiration [18]. To asses whether a similar cascade is activated by the mtCB_1_ receptor in adipocytes, we tested the ability of the sAC inhibitor KH7 and the PKA activator 8-Br-cAMP to block the cannabinoid effect on oxygen consumption [18,21].

Mitochondria were isolated from eWAT [30] of rats and underwent high-resolution respirometry analysis. Mitochondrial purification was validated by western blot, as indicated by the absence of plasma membrane, lysosome/endosome, and cytoplasm contamination [18,39,40] (Figure 3A). Interestingly, purified mitochondria contained CB_1_ protein and G alpha inhibitory protein (Gi), soluble adenylate cyclase (sAC), and the catalytic subunit of protein kinase A (PKA) (Figure 3B). As shown for mice homogenates, mitochondria purified from rat eWAT displayed proper responsiveness to complex I and complex II stimulation (Figure 3C).

WIN administration dose dependently reduced complex-I-dependent respiration (Figure 3D). This effect was still maintained upon DMSO pre-treatment but blunted by previous administration of the peripherally restricted CB_1_ receptor antagonist JD5037 (Figure 3E,F). Interestingly, as previously shown for brain-derived mitochondria [18], both KH7 and 8-br-cAMP pre-treatments were able to respectively occlude and prevent a WIN-induced decrease in complex-I-dependent respiration (Figure 3E,F).

This set of data suggests the involvement of the CB_1_-sAC-PKA pathway in cannabinoid’s impact on mitochondrial activity in white adipocytes.

## 4. Discussion

This study presents anatomical and functional evidence for the presence of CB_1_ receptor in WAT adipocyte mitochondria.

Conventional immunohistochemistry staining first allowed us to observe that the thin cytoplasmic rim of eWAT adipocytes contains a faint, albeit specific, CB_1_ receptor immunoreactivity, which was not observed in the iBAT samples. This goes in line with several pieces of evidence for consistent CB_1_ mRNA expression in eWAT despite the very low (even undetectable) presence in iBAT [33,34,35].

Ultrastructural morphometric analyses (i.e., immunogold staining coupled to electron microscopy) revealed the presence of CB_1_ receptors in several intracellular compartments, including mitochondria, of eWAT adipocytes. We thus investigated, in a second step, whether stimulation of the mitochondrial CB_1_ receptor pool negatively modulates mitochondrial activity. As shown for brain tissues [18,37], ex vivo administration of the cannabinoid agonist WIN reduced (i) complex I activity and (ii) oxygen consumption rate in eWAT homogenates. These latter effects are mainly due to the stimulation of CB_1_ receptors in adipocyte mitochondria since they are absent in *CB_1_*-KO [25], Ati-*CB_1_*-KO [16], and in DN22-*CB_1_*-KI mice [22]. The lack of effect of CB_1_ receptor activation on iBAT complex I activity might be explained by the fact that mitochondrial respiration in this tissue, as opposed to eWAT, is strongly uncoupled from ATP synthesis [41], thus leading to a possible impairment in the intramitochondrial cAMP-PKA pathway [41,42]. Indeed, this pathway represents the putative mechanism for mtCB_1_-dependent decrease in mitochondrial respiration (see [17,18,22]); under uncoupled conditions, the activation of the cAMP-PKA signaling inside mitochondria has no impact on mitochondrial activity [43,44,45], thus excluding any substrate for CB_1_ receptor-mediated mitochondrial alterations.

In isolated mitochondria from eWAT, CB_1_ receptors were detectable by western immunoblotting, together with Gi, sAC, and PKA catalytic subunit, with all proteins involved in the intramitochondrial mtCB_1_-dependent pathway previously described in the brain [18,43]. Indeed, stimulation of the receptor has been reported to cause a Gi protein-dependent inhibition of sAC, the enzyme responsible for the cAMP production, which hence controls PKA activity and the subsequent phosphorylation of complex I [18,22]. Since WIN reduced oxygen consumption in eWAT mitochondria, we assumed that this effect was mediated by mtCB_1_ receptor-dependent activation of similar pathways. Accordingly, by pre-treating mitochondria with JD5037 (CB_1_ antagonist [46]), KH7 (sAC inhibitor [43]), or 8-br-cAMP (PKA activator [47]), we were then able to block/occlude the WIN effects on mitochondrial respiration. In turn, these observations indicate that mitochondrial-associated CB_1_ receptors also engage a cAMP-PKA pathway to reduce respiration in white adipose tissues.

Starting from 2003, when CB_1_ receptor were first detected in adipocytes [48], multiple studies addressed the role of these receptors in peripheral organs, leading to the discovery that CB_1_ receptors directly control several physiological processes in many metabolically relevant tissues, including adipocytes [49,50]. The activity of the endocannabinoid system favors fat accumulation via central and peripheral mechanisms, including stimulation of adipocytes’ lipoprotein lipase (LPL) activity, fatty acid synthesis, triglyceride biosynthesis, and glucose entry [48,51], with all processes directly (or indirectly) controlled by mitochondrial metabolism [7]. Acute CB_1_ receptor agonism in cultured adipocytes and chronic receptor activation in WAT (e.g., upon obesity) have been shown to impair mitochondrial functions by decreasing mitochondrial biogenesis [13]. On the other hand, acute CB_1_ receptor blockade induces the expression of enzymes involved in the Krebs cycle, increasing the OXPHOS potential [52]. Accordingly, CB_1_ receptor antagonists affect complex IV activity and cellular oxygen consumption [53], thereby increasing energy expenditure in human and rodents suffering from obesity [52,54]. Altogether, this evidence points to mitochondrial metabolism as a likely target for CB_1_ receptor-mediated regulation of adipocyte physiology. Given that the localization of CB_1_ receptors is not limited to the plasma membrane but is found to be associated to the mitochondrial membranes in different brain cells [55,56], striatal muscle cells [19], spermatozoa, and oocytes [23,24], we unraveled a new subcellular pool of the receptor and its involvement in the regulation of mitochondrial functions [57].

PKA orchestrates several metabolic processes in adipocytes according to the nature of the fat depot involved. For example, sympathetic release of norepinephrine activates β3-adrenergic receptors on brown adipocytes, raising intracellular cAMP and PKA, and resulting in increased thermogenesis [58]. Furthermore, PKA activation in WAT enhances mitochondrial activity and can induce a brown-like phenotype in subcutaneous WAT depots [59,60]. Interestingly, the metabolic effects of peripheral CB_1_ blockade via pharmacological [46,61] or genetic tools (especially at the adipocyte level [16]) seem to be linked to higher mitochondrial metabolism following PKA-mediated activation of lipolysis [50,62]. This can provide the possible mechanism explaining the resistance to obesity and metabolic disorders observed upon impaired CB_1_ signaling at peripheral levels. Our observations that CB_1_ receptors directly control mitochondrial metabolism in eWAT perfectly fit within this *scenario* while also considering the subcellular targeting of PKA [63], a process mediated by A-kinase anchoring proteins (AKAPs), in several cell types, including adipocytes [64]. The mitochondria-localized AKAP1 tethers PKA to the cytosolic surface of the outer mitochondrial membrane in close proximity of local targets to maintain the mitochondrial function. In adipocytes, this phenomenon is responsible for the functional interaction between mitochondria and lipid droplets [65], a key process for lipolysis and fatty acid oxidation [66]. Thus, modulation of PKA activity by the mtCB_1_ receptor might not only underlie impairment in mitochondrial respiration but also result in perturbed AKAP-PKA trafficking, which might finally lead to altered substrate utilization in adipocytes, a phenomenon consistently observed in pathological conditions such as obesity and metabolic diseases [67].

In summary, the data presented in this study show that CB_1_ receptors are functionally expressed at the mitochondrial level in eWAT adipocytes; therein, they inhibit mitochondrial activity by directly engaging a sAC-PKA pathway inside the organelle. This discovery adds another mechanism through which endocannabinoids regulate adipocyte physiology and points out the possible involvement of mtCB_1_ receptors in several effects of CB_1_ receptor agonists and antagonists on adipose tissue, and hence in the regulation of energy metabolism. Furthermore, given the importance of peripheral CB_1_ receptors as key therapeutic targets for obesity and metabolic diseases [4], the characterization of adipose mtCB_1_ signaling in perturbed energy conditions may provide the base for developing efficient strategies for the treatment of obesity and related disorders. Targeting mtCB_1_ might thus help to avoid not only the serious neuropsychiatric side-effects of brain-penetrating CB_1_ receptor blockers but also potential issues related to the chronic blockade of the CB_1_ receptor in peripheral non-adipose tissues, promoting, e.g., gastrointestinal dysregulation, excessive sympathetic nervous system activity, and inflammatory processes.

## Figures and Tables

**Figure 1 cells-11-02582-f001:**
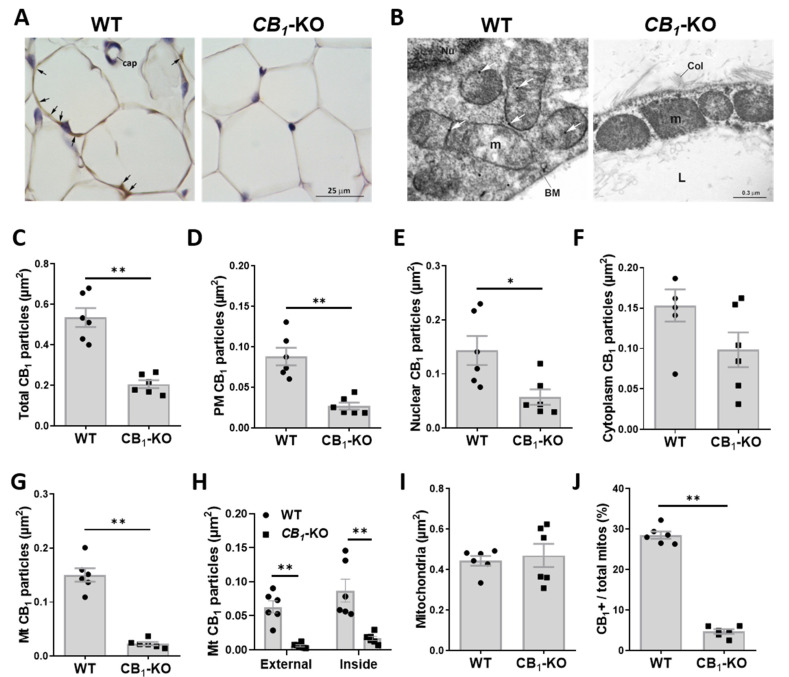
**Subcellular location of CB_1_ receptor in eWAT.** (**A**) Representative micrograph of CB_1_ receptor immunoreactivity in eWAT of WT vs. *CB_1_*-KO mice. Black arrows indicate specific CB_1_ receptor staining. Cap, capillary vessel. (**B**) Representative electron micrograph of immunogold detection of CB_1_ receptor (white arrows) in eWAT of WT vs. *CB_1_*-KO mice. BM, basal membrane; Col, collagen fiber; L, lipid droplet; m, mitochondria; Nu, nucleus. (**C**–**G**) Morphometric analysis and quantification of total CB_1_ receptor immunogold particles (**C**), plasma membrane (PM) CB_1_ receptor (**D**), nuclear CB_1_ receptor (**E**), cytoplasmic CB_1_ receptor (**F**), and mitochondrial-associated CB_1_ receptor (mtCB_1_) (**G**), normalized over the cytoplasm area in eWAT of WT vs. *CB_1_*-KO mice. (**H**) Morphometric quantification of CB_1_ receptor immunogold particles associated to mitochondrial membrane (Mt external; left) or inside mitochondria (Mt inside; right) over the cytoplasm area in eWAT of WT vs. *CB_1_*-KO mice. (**I**,**J**) Morphometric quantification of mitochondria number over the cytoplasm area (**I**) and relative percentage of CB_1_-positive mitochondria (**J**) in eWAT of WT vs. *CB_1_*-KO mice. * *p* < 0.05; ** *p* < 0.01.

**Figure 2 cells-11-02582-f002:**
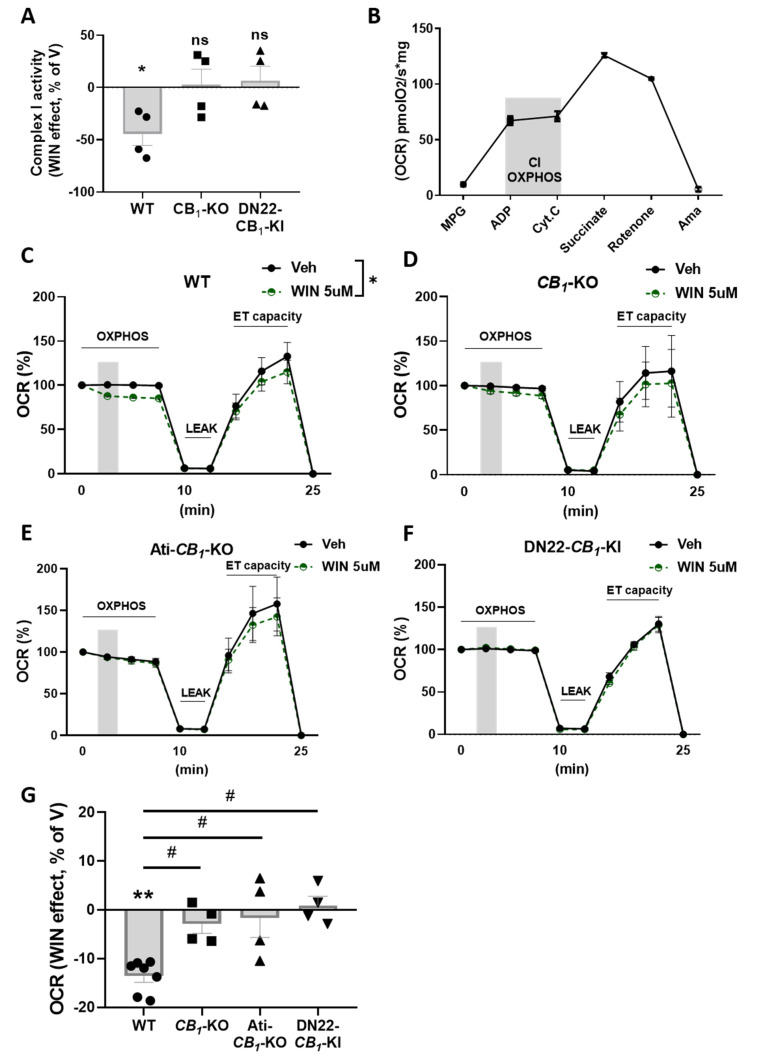
**Mitochondrial effects of CB_1_ activation in eWAT.** (**A**) WIN 5 µM effect (relative to vehicle, V) on complex I activity in eWAT from WT, *CB_1_*-KO, and DN22-CB1-KI mice. (**B**) Oxygen consumption rate (OCR) in eWAT homogenates from WT mice. MPG, malate pyruvate glutamate; ADP, adenosine di-phosphate; Cyt.C, cytochrome C; Ama, antimycin A. The gray area indicates complex-I-dependent respiration (OXPHOS) used for testing cannabinoid agonist effects. (**C**–**F**) Time course of the WIN effect on the oxygen consumption rate (OCR) in eWAT homogenates from WT (**C**), *CB_1_*-KO (**D**), Ati-*CB_1_*-KO (**E**), and DN22-*CB_1_*-KI (**F**) mice. The OXPHOS phase represents complex-I-dependent respiration (*see B*); LEAK, oligomycin-dependent ATP synthase inhibition; ETS, electron-transfer-system in an uncoupled state (by the addition of carbonyl cyanide m-chlorophenylhydrazone). The grey area indicates WIN (or vehicle) administration. (**G**) The WIN effect (relative to vehicle, V) on complex-I-dependent OXPHOS respiration in eWAT homogenates from WT, *CB_1_*-KO, Ati-*CB_1_*-KO, and DN22-*CB_1_*-KI mice calculated from panels (**C**–**F**), respectively. ** *p* < 0.01 from vehicle. # *p* < 0.05 from WT.

**Figure 3 cells-11-02582-f003:**
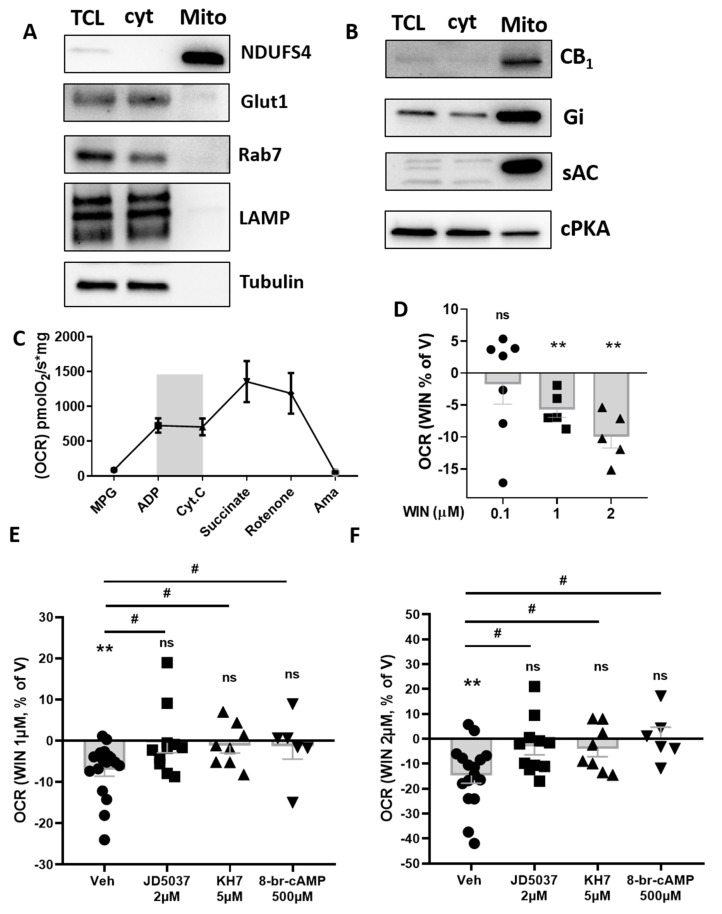
**Effect of CB_1_ stimulation and the relative intramitochondrial pathway involved in isolated eWAT mitochondria**. (**A**) Western blotting of isolated mitochondria extracts (mito) from rat eWAT compared to total cell lysate (TCL) and cytoplasmic (cyt) fraction. Membranes were incubated with antibodies against NADH:ubiquinone oxidoreductase subunit S4 (NDUFS4, mitochondrial marker), glucose tansporter-1 (Glut1, plasma membrane marker), Ras-related protein Rab-7a (Rab7, endosomes marker), lysosomal-associated membrane protein-2 (LAMP, lysosomes marker), and tubulin (cytoplasm marker). (**B**) Western blotting of isolated mitochondria extracts (mito) from rat eWAT compared to total cell lysate (TCL) and cytoplasmic (cyt) fraction. Membranes were incubated with antibodies against CB_1_ receptor, Gi protein, sAC (soluble adenylyl cyclase), and cPKA (catalytic subunit of protein kinase A). (**C**) Oxygen consumption rate (OCR) in isolated mitochondria from rat eWAT. MPG, malate pyruvate glutamate; ADP, adenosine di-phosphate; Cyt.C, cytochrome C; Ama, antimycin A. The gray area indicates complex-I-dependent respiration (OXPHOS) used for testing cannabinoid agonist effects. (**D**) Dose-dependent effect of WIN (relative to vehicle, V) on complex-I-dependent OXPHOS respiration in isolated mitochondria from rat eWAT. (**E**,**F**) Effect of WIN 1 µM (**E**) or 2 µM (**F**) (mean of 3 time points, expressed as a percentage of V) on complex-I-dependent OXPHOS respiration in isolated mitochondria from rat eWAT pretreated with control vehicle, 2 µM CB_1_ antagonist JD5037, 5 µM sAC blocker KH7, and 500 µM PKA activator 8-br-cAMP. * *p* < 0.05; ** *p* < 0.01 from vehicle. # *p* < 0.05 from Veh pre-treatment.

## Data Availability

All data are available from the corresponding author upon reasonable request.

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
