# Peer review of "Expression of Functional Cannabinoid Type-1 (CB1) Receptor in Mitochondria of White Adipocytes"

_cells, 2022, doi:10.3390/cells11162582_

Round 1

Reviewer 1 Report

The authors observed that CB1 is located at different subcellular levels, including plasma membrane and in close association with mitochondria. This founding is very important for understanding mechanism of function of cannabinoids.

Author Response

The authors observed that CB1 is located at different subcellular levels, including plasma membrane and in close association with mitochondria. This founding is very important for understanding mechanism of function of cannabinoids.

We are grateful to the Reviewer for his/her enthusiastic comments on our work.

Reviewer 2 Report

In this study the authors mainly showed that CB1 receptor is located at different subcellular levels, including plasma membrane and at the outer-membrane of mitochondria as well. Functional analysis was performed in tissue lysates and in isolated mitochondria to show that  CB1 receptor negatively contributes to complex I-dependent oxygen consumption in eWAT. By using pharmacological inhibitors and compounds, the authors also claimed that mtCB1 receptor-mediated activation involves the cAMP-PKA pathway. The current study provided some new insight into the action of cannabinoid in white adipocyte. However since the mitochondria-associated action of CB1 receptor has been reported in other tissues which significantly compromises the novelty of the current study. Moreover, the study design is too simple that does not yet reach the publication standard in the journal.

Major points:

1.  As pointed out by the authors themselves, mitochondrial-associated CB1 receptors have been reported in several other cell types like in brain, muscle, sperm and oocytes, the novelty of the current study is low. This is basically a repeat of previous studies in the eWAT, and therefore the results are largely expected.

2. In most of the studies to measure OCR, only relative level of OCR is shown (setting the untreated group as 100%). Are there any differences in OCR in different groups? For example, is there difference between OCR in WT and CB1 KO mitochondria? Will 8-br-cAMP alter OCR and metabolism of the mitochondria?

3. It is unclear why only eWAT was examined?

4. WAT is a heterogenous organ that contains adipocytes and other cell types such as macrophages. Is mt-associated CB1 receptor also observed in other cell types within the eWAT?

5. The introduction and discussion sections should be improved. Previous studies on mitochondria-associated CB1 receptor should be mentioned in more detail in the introduction or discussion part, since this is the major point of the study. The authors should focus more to convince the reader what is the advancement of the current study compared to the previous ones? Similarly, in the discussion section, the authors are suggested to talk less about the finding (no need to repeat the description of the result), but to elaborate more on the contribution of these findings in the field.

Minor points:

1. Line 348, similarly to mouse homogenate.

2. Line 356, Figure 3F instead of 4F.

Reviewer 3 Report

The present study by Pagano Zottola et al shows robust results on CB1 receptor localization and function in eWAT adipocyte mitochondria. CB1 activation reduced complex I-dependent oxygen consumption in eWAT, an effect that was not observed in CB1-KO, Ati-CB1-KO, and DN22-CB1-KO mice. Finally, the authors suggest that the reduction of mitochondrial respiration by mtCB1 activity is modulated by the intramitochondrial sAC-PKA pathway.

Minor comments:

To compare the impact of blocking mitochondrial respiration in DN22-CB1-KO mice, I miss the anatomical assessment of the presence (absence) of the CB1 receptor in the mitochondria of these mice.

Since the analysis of the sAC-PKA pathway was performed in rat eWAT, the authors could speculate on the effect of CB1-KO, Ati-CB1-KO and DN22-CB1-KO on the levels of this pathway and whether this putative effect may contribute to the reduction of mitochondrial respiration.

Author Response

The present study by Pagano Zottola et al shows robust results on CB1 receptor localization and function in eWAT adipocyte mitochondria. CB1 activation reduced complex I-dependent oxygen consumption in eWAT, an effect that was not observed in CB1-KO, Ati-CB1-KO, and DN22-CB1-KO mice. Finally, the authors suggest that the reduction of mitochondrial respiration by mtCB1 activity is modulated by the intramitochondrial sAC-PKA pathway.

We are grateful to the Reviewer for his/her enthusiastic comments on our work.

Minor comments:

To compare the impact of blocking mitochondrial respiration in DN22-CB1-KO mice, I miss the anatomical assessment of the presence (absence) of the CB1 receptor in the mitochondria of these mice.

We apologize to the reviewer for the lack of clarity. This mouse line was described in a previous paper (Soria-Gomez et al., Neuron 2021), in which we found that upon deletion of the first 22 amino acids, CB1 receptor is devoid of mitochondrial localization despite functional expression on plasma membrane. This paper is now more clearly cited in the new version of the manuscript.

Since the analysis of the sAC-PKA pathway was performed in rat eWAT, the authors could speculate on the effect of CB1-KO, Ati-CB1-KO and DN22-CB1-KO on the levels of this pathway and whether this putative effect may contribute to the reduction of mitochondrial respiration.

We added a new paragraph in the discussion (following also recommendation of Reviewer 1), focusing on the relation between CB1 receptor and PKA in adipose tissue.

Reviewer 4 Report

The study, "Expression of functional cannabinoid type-1 (CB1) receptor in mitochondria of white adipocytes" is a straight-forward observation by the authors that white adipocytes contain CB1 receptor. While this observation is well-substantiated, it is in and of itself insufficient for a self-contained publication. The authors have created a CB1 knockout, but there is no clear functional characterization of CB1 activity in WAT. The authors present data on modulation of CB1 impact on complex I activity, but looking at data in Figure 2C-E it appears that the only difference between the WT, CB1-KO, and Ati-CB1-KO may be due to changes in the error of the OCR(%) measurement. Otherwise these three plots appear to have nearly identical % change between Vehicle and treatment with 5 uM WIN. It is not apparent how the data in Figure 2C-E is reanalyzed to obtain the plot in Figure 2G. Thus, further analysis such as metabolite  profiling might be pursued, though it is unclear if a change between WT and CB1-KO might be observed. 

Thus, while the authors make a novel observation, this observation alone is insufficient to warrant publication.

Author Response

The study, "Expression of functional cannabinoid type-1 (CB1) receptor in mitochondria of white adipocytes" is a straight-forward observation by the authors that white adipocytes contain CB1 receptor. While this observation is well-substantiated, it is in and of itself insufficient for a self-contained publication. The authors have created a CB1 knockout, but there is no clear functional characterization of CB1 activity in WAT. The authors present data on modulation of CB1 impact on complex I activity, but looking at data in Figure 2C-E it appears that the only difference between the WT, CB1-KO, and Ati-CB1-KO may be due to changes in the error of the OCR(%) measurement. Otherwise these three plots appear to have nearly identical % change between Vehicle and treatment with 5 uM WIN. It is not apparent how the data in Figure 2C-E is reanalyzed to obtain the plot in Figure 2G. Thus, further analysis such as metabolite profiling might be pursued, though it is unclear if a change between WT and CB1-KO might be observed.

We apologize with the reviewer for the lack of clarity in the presentation of the figures. Figure 2G express the changes in OCR observed upon WIN administration, calculated in % of the changes observed in vehicle condition (average of the 3 time points). We improved the methods section to better describe how the data were represented.   

Thus, while the authors make a novel observation, this observation alone is insufficient to warrant publication.

We respectfully disagree with the Reviewer, since we propose a new mechanism trough which CB1 might alter mitochondrial activity in adipose tissue (e.g. by directly modulating these organelles). Given the key role that adipose mitochondria play in whole body energy balance regulation our study might provide a good example of a specific G-protein compartmentalization in the regulation of physiological (and pathological) processes.

Round 2

Reviewer 2 Report

The authors improved the manuscript to some extent. But I barely agree with the argument by the author that “we propose a new mechanism through which CB1 might alter mitochondrial activity in adipose tissue (e.g. by directly modulating these organelles).”, because this mechanism has been proposed and evidenced by a previous work (Nature volume 539, pages555–559 (2016)), the only difference is in different tissues. They’ve already shown that mtCB1 receptors signal through intra-mitochondrial Gαi protein activation and consequent inhibition of soluble-adenylyl cyclase (sAC). I agree with the author that the current work is performed in different tissues, i.e., the context is different and therefore still blessed with certain novelty at certain extent. But the work in the current form is still over-simplified for publication in the journal and more in-depth study is needed.

1. Whether phosphorylation of specific subunits of the mitochondrial complex (such as complex I and II) is altered upon agonism of mtCB1 receptor-Gi-sAC-PKA axis in white adipocyte mitochondria? This has been proved in (Nature volume 539, pages555–559 (2016)) but would still be interesting to look into in the adipocytes. The result will also strengthen the current work.

2. In addition to the previously reported mechanism of mtCB1 receptor, it is worthwhile for the author to further check whether modulation of PKA activity by mtCB1 receptor result in a perturbed AKAP-PKA trafficking, which will be a truly novel point for the current study.

3. A co-immunofluorescent staining of macrophage marker (such as F4/80) is needed to strengthen the point that CB1 receptor is not expressed in adipose macrophages.

Author Response

The authors improved the manuscript to some extent. But I barely agree with the argument by the author that “we propose a new mechanism through which CB1 might alter mitochondrial activity in adipose tissue (e.g. by directly modulating these organelles).”, because this mechanism has been proposed and evidenced by a previous work (Nature volume 539, pages555–559 (2016)), the only difference is in different tissues. They’ve already shown that mtCB1 receptors signal through intra-mitochondrial Gαi protein activation and consequent inhibition of soluble-adenylyl cyclase (sAC). I agree with the author that the current work is performed in different tissues, i.e., the context is different and therefore still blessed with certain novelty at certain extent. But the work in the current form is still over-simplified for publication in the journal and more in-depth study is needed.

We thank the Reviewer for his/her constructive comments on our work. However we still respectfully disagree with his/her considerations about the novelty of our work, which in our opinions is indeed the cannabinoid modulation of mitochondrial activity in white adipose tissue.

  1. Whether phosphorylation of specific subunits of the mitochondrial complex (such as complex I and II) is altered upon agonism of mtCB1 receptor-Gi-sAC-PKA axis in white adipocyte mitochondria? This has been proved in (Nature volume 539, pages555–559 (2016)) but would still be interesting to look into in the adipocytes. The result will also strengthen the current work.

This is a very interesting point and indeed our experimental data obtained in eWAT isolated mitochondria point in this direction (see last comments from Reviewer 3). However, given the fact that we are currently in the August summer break of our research center, it would be frankly impossible to perform the series of studies proposed by the Reviewer, since there isn’t any possibility to obtain animals and/or reagent and antibodies in the timeframe of 10 days. Furthermore, such experiments will require a series of preliminary tests which we are not able to perform in the timeframe of 10 days.

Regarding the issue whether mtCB1 modulates PKA-dependent phosphorylation of ComplexI or ComplexII, we are quite convinced from previous studies that ComplexI might be the good candidate for 2 reasons. 1) The inhibitory effect of CB1 agonists on mitochondrial activity is present upon ComplexI stimulation and not ComplexII. 2) Expression of phospho-mimetic subunits of ComplexI (NDUFS2 and NDUFS4) for PKA target sites is sufficient to block mitochondrial, cellular and behavioral effects of mtCB1 activation.

  1. In addition to the previously reported mechanism of mtCB1 receptor, it is worthwhile for the author to further check whether modulation of PKA activity by mtCB1 receptor result in a perturbed AKAP-PKA trafficking, which will be a truly novel point for the current study.

This is a very interesting and novel point. However, such experiments will require a series of preliminary tests (including a complete new setup of adipocyte cell models) which we are not able to perform in the timeframe of 10 days, given also the August summer break. Furthermore, we are sincerely convinced that these studies, although very interesting, are beyond the scope of the present manuscript.

  1. A co-immunofluorescent staining of macrophage marker (such as F4/80) is needed to strengthen the point that CB1 receptor is not expressed in adipose macrophages.

We agree with the Reviewer on this point however, we are quite convinced that at least functionally there is no CB1 receptor on adipose macrophages since Ati-CB1-KO mice are unresponsive to the effect of CB1 activation on mitochondrial respiration. Moreover, we will not be able to perform such experiment in the 10 days timeframe for the reasons mentioned above.

Reviewer 4 Report

In the revised manuscript: "Expression of Functional Cannabinoid Type-1 (CB1) Receptor in Mitochondria of White Adipocytes" the authors employ a series of cellular studies to look at the localization of CB1 in eWAT. They then explore the impact of CB1 on mitochondrial respiration.

I appreciate the authors clarification of Figure 2G. With this improved clarity I have a number of additional points that I think need to be addressed prior to acceptance.

Section 3.1 and 3.2: The studies looking at CB1 localization are well conducted and clearly show CB1 localization in plasma membrane, nucleus, and mitochondria. It therefore seems that the title of 3.2 is somewhat misleading as it implies that CB1 receptor is uniquely associated with eWAT mitochondria, while their data clearly shows that similar association could be seen for both plasma membrane and nucleus. 

3.3: This section is much improved, especially with the clarification of the DN22-CB1-KI mice phenotype. I also appreciated the improved description of the methods. 

3.4: In 3.4, the authors utilize a series of inhibitors to suggest involvement of the sAC-PKA pathways in CB1 inhibitor activity. The comparison of data in 3D and 3E/F lane1 is somewhat concerning as it seems that there is a significant change in the effect of WIN in the combination experiment that is used to try and demonstrate KH7 and 8-br-cAMP activity. The authors for this appear to be relying on the separation of Vehicle from control rather than performing a comparator test to determine if the co-treatment is different from the WIN/Veh control. I believe that such an analysis might not prove to be statistically different and thus would negate the authors statement. 

Finally, I believe that the authors overstate the ultimate outcome of the study. In section 3.4, the authors state that their data "suggests involvement of a CB1-sAC-PKA" pathway, yet in the discussion, they state that the data presented shows that "CB1 receptors ... inhibit mitochondrial activity by directly engaging a sAC-PKA pathway inside the organelle". While the presented data shows mitochondrial localization, localization inside mitochondria, and activity at mitochondria. It does not rise to the level of demonstrating direct engagement of the pathway. Especially as the data in Fig 3E and F is only marginally convincing that the sAC and PKA pathways are involved. I suggest that the authors limit the concreteness of their outcome.

- minor point: please ensure consistent use of eWAT throughout, there are a couple places where epididymal WAT is maintained (e.g. 311-312)

Author Response

In the revised manuscript: "Expression of Functional Cannabinoid Type-1 (CB1) Receptor in Mitochondria of White Adipocytes" the authors employ a series of cellular studies to look at the localization of CB1 in eWAT. They then explore the impact of CB1 on mitochondrial respiration.

I appreciate the authors clarification of Figure 2G. With this improved clarity I have a number of additional points that I think need to be addressed prior to acceptance.

We thank the Reviewer for his/her nice and constructive comments on our work

Section 3.1 and 3.2: The studies looking at CB1 localization are well conducted and clearly show CB1 localization in plasma membrane, nucleus, and mitochondria. It therefore seems that the title of 3.2 is somewhat misleading as it implies that CB1 receptor is uniquely associated with eWAT mitochondria, while their data clearly shows that similar association could be seen for both plasma membrane and nucleus.

We agree with the Reviewer and we changed the session title.

3.3: This section is much improved, especially with the clarification of the DN22-CB1-KI mice phenotype. I also appreciated the improved description of the methods.

We thank the Reviewer for helping us clarifying our models’ presentation

3.4: In 3.4, the authors utilize a series of inhibitors to suggest involvement of the sAC-PKA pathways in CB1 inhibitor activity. The comparison of data in 3D and 3E/F lane1 is somewhat concerning as it seems that there is a significant change in the effect of WIN in the combination experiment that is used to try and demonstrate KH7 and 8-br-cAMP activity. The authors for this appear to be relying on the separation of Vehicle from control rather than performing a comparator test to determine if the co-treatment is different from the WIN/Veh control. I believe that such an analysis might not prove to be statistically different and thus would negate the authors’ statement.

We apologize with the Reviewer for not showing these comparisons which are actually statistically significant. We now modified Figure 3 and Figure S2 according to Reviewer request

Finally, I believe that the authors overstate the ultimate outcome of the study. In section 3.4, the authors state that their data "suggests involvement of a CB1-sAC-PKA" pathway, yet in the discussion, they state that the data presented shows that "CB1 receptors ... inhibit mitochondrial activity by directly engaging a sAC-PKA pathway inside the organelle". While the presented data shows mitochondrial localization, localization inside mitochondria, and activity at mitochondria. It does not rise to the level of demonstrating direct engagement of the pathway. Especially as the data in Fig 3E and F is only marginally convincing that the sAC and PKA pathways are involved. I suggest that the authors limit the concreteness of their outcome.

See previous point, with our new statistical analysis we are more convinced about the concreteness of our data showing the involvement of sAC-PKA pathway.

- minor point: please ensure consistent use of eWAT throughout, there are a couple places where epididymal WAT is maintained (e.g. 311-312)

We changed the text accordingly.